

# Systematic mappings on technology adoption in small and medium-sized enterprises from developing countries: an umbrella review

Jaime Díaz-Arancibia[1], Ana Bustamante-Mora[1],
Jorge Hochstetter-Diez[1] and Gabriel Mauricio Ramírez Villegas[2]

[1] Departamento de Ciencias de la Computación e Informática, Universidad de La Frontera, Temuco, La Araucanía, Chile
[2] Facultad de Ingenierías, Universidad de Medellin, Medellín, Colombia

Corresponding author
Jaime Díaz-Arancibia,
jaimeignacio.diaz@ufrontera.cl

## ABSTRACT

This article synthesizes insights from 21 systematic mappings examining how small and medium-sized enterprises (SMEs) in developing countries adopt and benefit from digital transformation. The review identifies recurrent barriers such as managerial resistance, limited digital literacy, and fragmented policies by integrating diverse literature sources. These challenges underscore the importance of crafting region-specific digital frameworks and strengthening public-private collaborations to bolster SMEs' technological capabilities. Findings highlight that foundational tools (*e.g.*, social media and cloud computing) provide immediate value in customer engagement and resource management. Meanwhile, advanced technologies—such as artificial intelligence (AI), blockchain, and the Internet of Things (IoT)—exhibit transformative potential by enhancing operational efficiency and transparency, particularly in supply chain and data-driven processes. However, structural constraints, including inadequate infrastructure and uneven policy support, impede broader adoption. Established models like the Technology Acceptance Model (TAM) and the Technology-Organization-Environment (TOE) framework prove partially effective but often fail to capture socio-cultural complexities prevalent in resource-constrained environments. Consequently, this article advocates adaptive strategies that integrate cultural nuances and collaborative ecosystems linking governments, industry, and academic institutions. The study concludes by outlining specific avenues for future research—such as deeper explorations of socio-cultural integration in technology models and developing context-sensitive policy frameworks—ultimately fostering long-term digital strategies that enable sustainability, resilience, and inclusive growth for SMEs in emerging markets.

## INTRODUCTION

Digital transformation has become a cornerstone for enhancing the competitiveness, sustainability, and operational efficiency of small and medium-sized enterprises (SMEs). In many developing countries, these organizations face particularly acute challenges

(*Kahveci et al., 2025*), including limited financial resources, inadequate technological infrastructure, and insufficient digital literacy (*Alhamami et al., 2021*; *Pangarso et al., 2022a*; *Pratama, Santoso & Mustaniroh, 2021*; *Sinha, Raby & Salari, 2025*). However, the rapid advancement of innovations such as artificial intelligence (AI), blockchain, and cloud computing could revolutionize industries if appropriate strategies are devised to overcome resource constraints (*Viloria, Iranmanesh & Grybauskas, 2022*; *Alsibhawi, Yahaya & Mohamed, 2023*; *Ragazou, Tsami & Nikitakos, 2022*). Over the past decade, the widespread adoption of Information and Communication Technologies (ICTs) has further accelerated digitalization across sectors, offering avenues to improve operational efficiency and fuel innovation (*Vial, 2021*). This trend is particularly relevant to SMEs, which often struggle to compete globally but stand to benefit substantially from digital initiatives (*Kaplan & Haenlein, 2019*).

Despite growing work on SME digitalization, systematic mappings focusing on resource-limited contexts remain sparse. Existing theoretical models like the Technology Acceptance Model (TAM) and the Technology-Organization-Environment (TOE) framework have predominantly been validated in larger organizations or developed economies, frequently overlooking the sociocultural and infrastructural complexities of emerging markets (*Manaf et al., 2022*; *Hofstede, 2001*; *Cruz & Esteban, 2019*). This knowledge gap is significant for various stakeholders, including academics seeking to refine technology adoption theories, policymakers formulating supportive measures for SME development, and practitioners aiming to implement digital solutions on the ground. Our previous studies by *Díaz-Arancibia et al. (2024)*, and others underscore that many of the barriers and enablers for digital transformation in these environments differ markedly from those observed in high-income regions, indicating a pressing need for more context-specific analysis. For clarity, we use *barriers* to denote organizational, technological, and environmental constraints that inhibit the initiation or scaling of digital tools (*e.g.*, managerial resistance, skills gaps, infrastructure deficits), and *drivers/enablers* to refer to conditions and capabilities that facilitate adoption and deepening (*e.g.*, leadership support, targeted training, public incentives, ecosystem partnerships).

This article responds to that need by consolidating insights from systematic mappings, thus offering a current and comprehensive overview of how SMEs in developing countries integrate digital technologies under constrained conditions. Beyond European contexts, we explicitly incorporate evidence from Latin America, Africa, and Asia to ground the synthesis in developing-country realities (*e.g.*, *Costa Júnior et al., 2022*; *Hossain, Akhter & Sultana, 2022*; *Irawan et al., 2022*; *Olokundun et al., 2022*; *Ammeran, Noor & Yusof, 2023*).

The review examines the practical relevance of emerging tools (artificial intelligence (AI), IoT, blockchain) for these smaller enterprises, highlighting where existing frameworks fall short and suggesting adaptations that integrate local norms, infrastructure realities, and cultural factors. Such a synthesis is necessary given the fragmented state of research, where previous reviews have often focused narrowly on single technologies or specific barriers without delivering an aggregated perspective that captures the interplay of

organizational, environmental, and human elements (*Alhamami et al., 2021*; *Pangarso et al., 2022a*). By gathering these dimensions into one holistic analysis, the article aims to help readers from diverse fields—such as ICT4D, development economics, and technology management—grasp the critical factors determining successful digital adoption and scale-up in SMEs across emerging economies.

In the final sections, we address how established models might be tailored or combined with newer approaches that consider social contexts and resource constraints, thereby enhancing theoretical robustness and practical feasibility. We also discuss the challenges related to policy fragmentation and financing gaps, proposing collaboration between government, industry, and academia to strengthen digital ecosystems (*Viloria, Iranmanesh & Grybauskas, 2022*). The structure of the article is as follows: After detailing the methodology used to conduct this systematic review, we present and discuss the results, emphasizing identified barriers, proposed solutions, and potential research avenues. We then conclude with broader implications, limitations, and suggestions for future work, underscoring the importance of cross-sector efforts in driving sustainable and resilient digital transformation in SMEs from developing regions.

## BACKGROUND

### SMEs in resource-constrained environments

Small and medium enterprises (SMEs) are regarded as vital engines of economic growth, employment generation, and innovation (*OECD, 2022*; *World Bank, 2021*). Despite varying country-specific definitions, SMEs commonly have an upper limit of employees—often between 50 and 250—along with thresholds for annual turnover or assets (*OECD, 2022*). The World Bank, for instance, typically classifies micro-enterprises as those with fewer than 10 employees, small enterprises as those with 10 to 50 employees, and medium enterprises as those with up to 300 employees (*World Bank, 2021*). In developing countries, these cutoffs can be fluid, reflecting the informal nature of many businesses. Regardless of the definition, SMEs collectively represent a substantial share of economic activity and job creation (*UNIDO, 2020*).

However, SMEs operating in resource-constrained environments face additional hurdles: limited capital availability, technology gaps, and lack of specialized human resources (*Alnafrah & Alharbi, 2020*). These constraints often compound the broader challenges of digital transformation, making examining how SME-specific strategies differ from those employed by larger organizations with more abundant resources critical. We adopt the PICOC scheme—Population, Intervention, Comparison, Outcome, and Context—to structure the scope and the search logic of this umbrella review. In our case: **Population** = micro, small, and medium-sized enterprises (SMEs) in developing countries; **Intervention** = digital transformation and technology-adoption frameworks; **Comparison** = not applicable (umbrella review without a control arm); **Outcome** = actionable, transferable insights for scalable adoption; and **Context** = low-resource settings characterized by infrastructural and financial constraints. Framing the study in this way ensures that both inclusion criteria and evidence synthesis remain aligned with the realities of SMEs operating in resource-constrained environments.

## General concepts

### Digital transformation and technology adoption

Digital transformation is a multidisciplinary process that leverages digital technologies to improve organizational strategies, enhance employee capabilities, and modernize technical infrastructures (*Cavalcanti, Oliveira & de Oliveira Santini, 2022*). This extends beyond merely digitizing existing processes, demanding shifts in culture, leadership, and managerial mindset (*Vial, 2021*). In this context, *technology adoption* refers to the decision-making process through which new technologies are selected, introduced, and integrated into operational workflows. Such adoption is influenced by various factors, including organizational readiness, user perceptions, and cultural dynamics (*Kaplan & Haenlein, 2019*; *Kane et al., 2019*).

An important concept when discussing digitalization in developing regions is the *digital divide* or *gap*, understood as the discrepancy between current levels of digitalization (*e.g.*, internet penetration, availability of e-services) and what could be achieved under optimal conditions. This gap becomes more pronounced for SMEs in resource-limited settings, as inadequate infrastructure, limited training opportunities, and high costs can exacerbate the difference between existing usage and potential technological integration (*Alnafrah & Alharbi, 2020*).

**Firm-size strata and implications for adoption.** Consistent with international practice, we distinguish SME size classes by *headcount*—micro (1–9 employees), small (10–49), and medium (50–249)—which are more stable across jurisdictions than revenue bands (*OECD, 2022*). For developing-country contexts, we also report *illustrative* annual sales anchors drawn from SME finance work to provide an economic sense of scale, noting that legal thresholds vary by country (*UNIDO, 2020*; *World Bank, 2021*). Microenterprises (with ≤10 workers and/or typical sales below ~$100,000) tend to operate with informal structures, self-financing, and a strong dependence on the owner–manager for decision-making. **Small firms** (10–49 workers and/or sales roughly between ~$100,000 and ~$1,050,000) usually exhibit partial formalization, some managerial specialization, and episodic access to external finance and IT services. In contrast, **medium-sized enterprises** (50–249 employees and/or sales at or above ~$1,050,000) more often have professional management teams, internal IT capacity, and broader access to capital markets. These gradations in organizational resources and governance condition both the *speed* and *depth* of technology adoption, and they motivate our configurational lens that combines internal endowments with ecosystem density in the remainder of the article.

## Developing countries

Developing countries exhibit structural challenges stemming from lower socioeconomic development, limited technological infrastructure, and complex regulatory frameworks (*Alnafrah & Alharbi, 2020*). The World Bank often classifies these nations based on income levels (low, lower-middle, or upper-middle) alongside human development indicators such as literacy rates and access to essential services (*World Bank, 2021*). Digital transformation initiatives within these regions can expedite broader development goals

like inclusive healthcare, education, and financial inclusion (*Global Education Monitoring Report Team, 2023*; *Tallafokam & Mbolela, 2023*). However, local cultural norms and diverse governance mechanisms often necessitate tailored solutions (*Hofstede, 2001*).

A central concern is ensuring digital technologies do not inadvertently widen social inequities. For instance, adopting advanced platforms without adequate training may benefit only a segment of the population—those who are already digitally literate—thus deepening divides between privileged and vulnerable groups (*Rahayu & Day, 2017*; *Lestari & Sensuse, 2021*). Hence, equity and inclusion must be at the forefront of digital transformation policies, ensuring that marginalized communities and smaller enterprises can participate fully in the digital economy.

Empirical studies from Brazil, Bangladesh, Indonesia, and Nigeria, among others, consistently report binding constraints in finance, digital skills, and supplier sophistication, and document how SMEs routinize coping practices under scarcity (*Costa Júnior et al., 2022*; *Hossain, Akhter & Sultana, 2022*; *Irawan et al., 2022*; *Olokundun et al., 2022*). These patterns align with our umbrella findings on barrier families and help contextualize the ecosystem-level mechanisms discussed later.

Umbrella reviews occupy the apex of the evidence hierarchy because they synthesize patterns that individual mappings cannot reveal (*Post et al., 2020*). Following *Whetten*'s *(1989)* "What–How–Why–Who–Where" logic, our review introduces a Meso-level construct—*ecosystem density*—to explain *how* internal resources translate (or fail to translate) into deep technology adoption *why* through barrier mediation, and delineates *who/where* (resource-constrained SMEs in developing countries) these relationships apply.

## TAM and TOE frameworks

The Technology Acceptance Model (TAM) and the Technology-Organization-Environment (TOE) framework have been instrumental in understanding how organizations perceive and adopt new technologies (*Davis, 1989*; *Tornatzky & Fleischer, 1990*). TAM posits that perceived ease of use and usefulness are primary determinants of user acceptance, focusing heavily on individual or user-level factors (*Ramdani, Kawalek & Lorenzo, 2009*; *Jia & Ding, 2017*). Conversely, TOE incorporates organizational and environmental elements, thus providing a more holistic view of readiness and external pressures (*Ramdani, Kawalek & Lorenzo, 2009*; *Tornatzky & Fleischer, 1990*).

Nevertheless, these models often overlook the distinctive constraints in developing economies, particularly infrastructure, policy fragmentation, and cultural dynamics (*Kaplan & Haenlein, 2019*; *Marcus & Gould, 2001*). Critics argue that frameworks initially validated in high-income or large-scale corporate contexts cannot be uncritically applied to SMEs in resource-limited environments (*Cruz & Esteban, 2019*). As such, researchers increasingly advocate for adaptations that integrate community-driven innovation, local cultural practices, informal financing structures, and minimal technical support (*Kaplan & Haenlein, 2019*; *Ragazou, Tsami & Nikitakos, 2022*). These adaptations become pivotal in accurately capturing the interplay between SME characteristics, resource constraints, and the sociocultural backdrop where digital transformation efforts unfold.

### Measuring the "gap"

While the term *gap* frequently appears in policy and academic literature, its precise measurement varies. Some studies quantify it through internet accessibility or bandwidth disparities; others use metrics like e-commerce uptake, adoption rates of cloud services, or the ratio of digital tools employed relative to business size (*Rahayu & Day, 2017*). In practical terms, identifying a gap often involves comparing current technology adoption levels against a benchmark of ideal or widespread usage. Therefore, when we refer to barriers or gaps, we imply an observed shortfall from these expected or optimal adoption levels—not merely an abstract difference. Consequently, closing this gap entails strategies designed to bring SMEs' technological capabilities closer to that benchmark, whether it be through training, financial incentives, or collaborative innovation networks.

Beyond firm-internal factors, adoption outcomes in developing countries depend on meso-level conditions; What the entrepreneurial ecosystem literature describes as the density and quality of networks, specialized suppliers, finance, and support organizations. We therefore treat "ecosystem density" as a contextual moderator that shapes how internal resources translate into adoption depth (cf. *Isenberg, 2010*; *Stam, 2015*; *Spigel, 2017*).

**Rationale for an umbrella review.** Despite the growing number of systematic mappings on digital transformation in SMEs, the resulting evidence base remains *fragmented*: each mapping employs different inclusion criteria, analytical lenses, and definitions of "technology adoption". Consequently, cross-study comparisons are difficult and the cumulative knowledge required for robust theory-building and policy design is missing. This fragmentation constitutes the core research gap: we lack an overarching synthesis that can (i) collate disparate findings, (ii) reconcile inconsistent operationalisations of "adoption gaps", and (iii) reveal higher-level patterns across regions, sectors, and methodological traditions. Following the evidence-synthesis hierarchy from *Pollock (2017)*, an *umbrella review—i.e.*, a review of systematic reviews and mappings—provides the appropriate methodological vehicle. It aggregates secondary studies, appraises overlap and methodological quality, and generates meta-level insights that are not visible within single mappings. Our review therefore addresses the above gap by offering a consolidated, quality-assessed view of technology-adoption research in SMEs from developing countries.

## METHODS

### Overview of systematic mapping studies

Systematic mapping studies (SMS) provide a structured approach to mapping broad research fields, commonly used in software engineering and multidisciplinary domains. By adopting a predefined scope and a series of rigorous steps, SMS can reveal trends, research gaps, and clusters of scholarly activity (*Petersen et al., 2015*).

To uphold both rigor and relevance, our selection and evaluation procedures were grounded in the methodological framework proposed by *Petersen et al. (2015)*, complemented by adjustments tailored to the specific nature of this investigation. We formulated our research questions through an iterative process: first, a preliminary literature scan helped us identify principal themes and gaps; subsequently, consultations
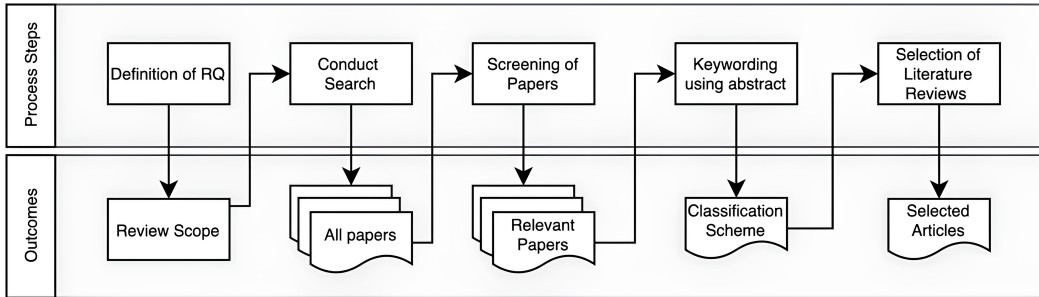

**Figure 1 The main phases of the SMS process.** Adapted from *Díaz-Arancibia et al. (2024)*.

with field experts refined these questions to represent better the core challenges of digital transformation in developing countries. Because our analysis is limited exclusively to secondary studies (*i.e.*, literature reviews and systematic mappings), the systematic mapping study (SMS) approach is particularly well-suited to exploring how digital technologies are adopted within micro and small economies. By employing this structured methodology, we aim to offer a detailed perspective on how digital innovations are integrated into these contexts, ultimately shaping a more informed and nuanced research agenda. Figure 1 illustrates the main phases of the SMS process, where each phase informs subsequent steps.

**An umbrella review.** Traditional systematic mappings yield valuable but isolated snapshots. To move the field forward, higher-order synthesis is required. We therefore adopt an umbrella-review design that (a) collates existing secondary studies, (b) assesses inter-review consistency, and (c) surfaces meta-patterns and residual evidence gaps. This design choice is consistent with published guidance on evidence hierarchies for complex interventions in management and information-systems research (*Pollock, 2017*; *Petersen et al., 2015*).

## Defining the research questions

Defining the research questions (RQs) marks the project's starting point, establishing the methodological framework for the subsequent steps. Table 1 lists the four RQs guiding this study, and Fig. 2 illustrates the focal domains for each: (a) Digital Transformation Evaluation Models for SMEs, (b) Socio-Cultural Behavior, and (c) Drivers and Barriers. The convergence of these three aspects is highlighted in (d). The RQs align with the study's objectives, examining key frameworks, sociocultural behaviors, and barriers/enablers in SMEs within developing countries. Figure 2 visualizes how each RQ maps to specific thematic areas.

## Search strategy

Following the PICOC (Population, Intervention, Comparison, Outcome, Context) framework, we constructed our search terms by combining synonyms (using "OR") and distinct concepts (using "AND"). Table 2 summarizes the PICOC criteria. We focused on

**Table 1  Research questions.** Adapted from *Díaz-Arancibia et al. (2024)*.

| ID | Item | Objective |
|---|---|---|
| RQ1 | *(a) Which digital transformation models support the evaluation of technological adoption in SMEs?* | To identify general technology adoption models. |
| RQ2 | *(b) What are the mechanisms for assessing cultural behavior in micro and small enterprises?* | To identify formal models on cultural behavior and socio-cultural impact on digital transformation. |
| RQ3 | *(c) What are the technological barriers affecting SMEs in developing countries?* | To identify technology gaps and barriers. |
| RQ4 | *(c) What are the determining factors and drivers in the digital transformation process for SMEs in developing countries?* | To identify drivers to the digital transformation process. |

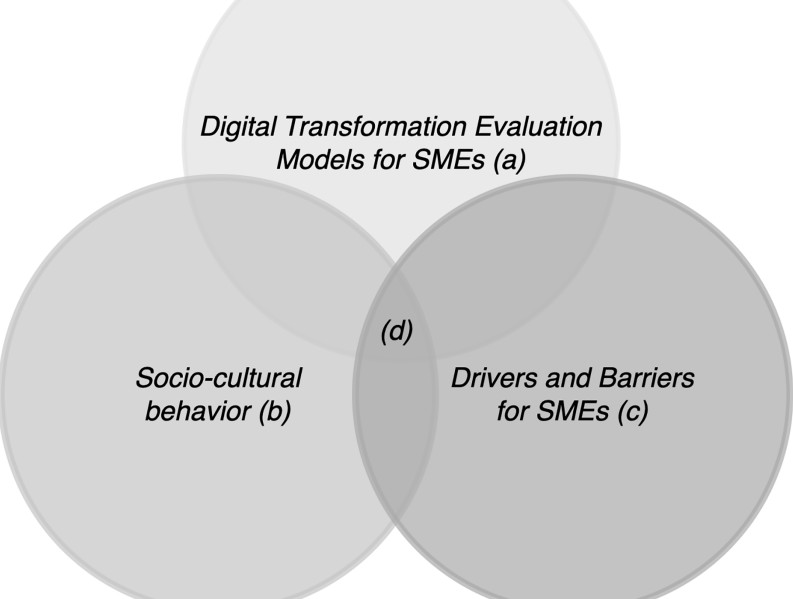

**Figure 2  Mapping of research questions to thematic areas.** Adapted from *Díaz-Arancibia et al. (2024)*.

**Table 2  PICOC criteria.** Adapted from *Díaz-Arancibia et al. (2024)*.

| Criteria | Definitions |
|---|---|
| *Population* | Micro enterprise, small enterprise, medium enterprise |
| *Intervention* | Technology adoption model, digital transformation |
| *Comparison* | n/a |
| *Outcome* | Process, initiatives, proposals, products, software, model |
| *Context* | Academy, industry |

**Table 3 Keywords and synonyms.** Adapted from *Díaz-Arancibia et al. (2024)*.

| Keyword | Synonyms |
|---|---|
| *Digital transformation* | Digital disruption, digital innovation, digital modernization, digitalization |
| *Model* | Approach, practice, procedure, protocol, technique, framework, initiative, method, process, product, proposal, software |
| *Small enterprise* | SME, SMEs, medium enterprise, micro enterprise |
| *Technology adoption model* | TAM, TOE, technology acceptance, technology adaptation, Tornatzky |

**Table 4 Research chain.** Replicated from *Díaz-Arancibia et al. (2024)*.

**Research chain**

("small enterprise" OR "SME" OR "SMEs" OR "medium enterprise" OR "micro enterprise") AND ("digital transformation" OR "Digital disruption" OR "Digital innovation" OR "Digital modernization" OR "digitalization" OR "technology adoption model" OR "TAM" OR "TOE" OR "Technology acceptance" OR "Technology adaptation" OR "Tornatzky") AND

("model" OR "Approach" OR "Practice" OR "Procedure" OR "Protocol" OR "Technique" OR "framework" OR "initiative" OR "method" OR "process" OR "product" OR "proposal" OR "software")

SMEs (Population), exploring digital transformation frameworks (Intervention) to identify actionable insights (Outcome) within resource-constrained environments (Context).

**Geographic scope (clarification).** Since our research question focuses on SMEs operating in developing-country contexts, we did not apply *ex ante* geographic filters to outlets. Restricting the *corpus* to any single region (*e.g.*, Europe) could bias the evidence base. After conducting an exercise based on European outlets, no additional *secondary* studies met our inclusion/exclusion criteria. We therefore retain the original *corpus* while explicitly reporting this verification step.

Table 3 outlines the process for constructing the search string. Groups of equivalent terms are linked *via* the "OR" operator, whereas different thematic domains are joined by "AND." Although minor adjustments to the query may be required depending on the search engine, these do not alter the study's overall findings.

Taking into account the various requirements and criteria, the final search string was determined as follows (see Table 4):

### Sources and data collection

Relevant literature was extracted from academic databases, including ACM, IEEE, Scopus, Web of Science, and Wiley. An iterative filtering process (Fig. 3) was used to apply our inclusion and exclusion criteria rigorously. In line with guidance on incorporating indexed, protocol-transparent grey literature in management/SE syntheses, we retained conference/book–series reviews that met indexing and method transparency criteria; outlet-type robustness is demonstrated in Methods (Robustness to outlet type and evidence stratification). Figure 4 describes the analytical pipeline for the umbrella review.

### Article selection

Studies explicitly describing a replicable search and selection protocol (systematic review, systematic mapping, or scoping review) were eligible; empirical primary studies and purely discursive conceptual articles were excluded.

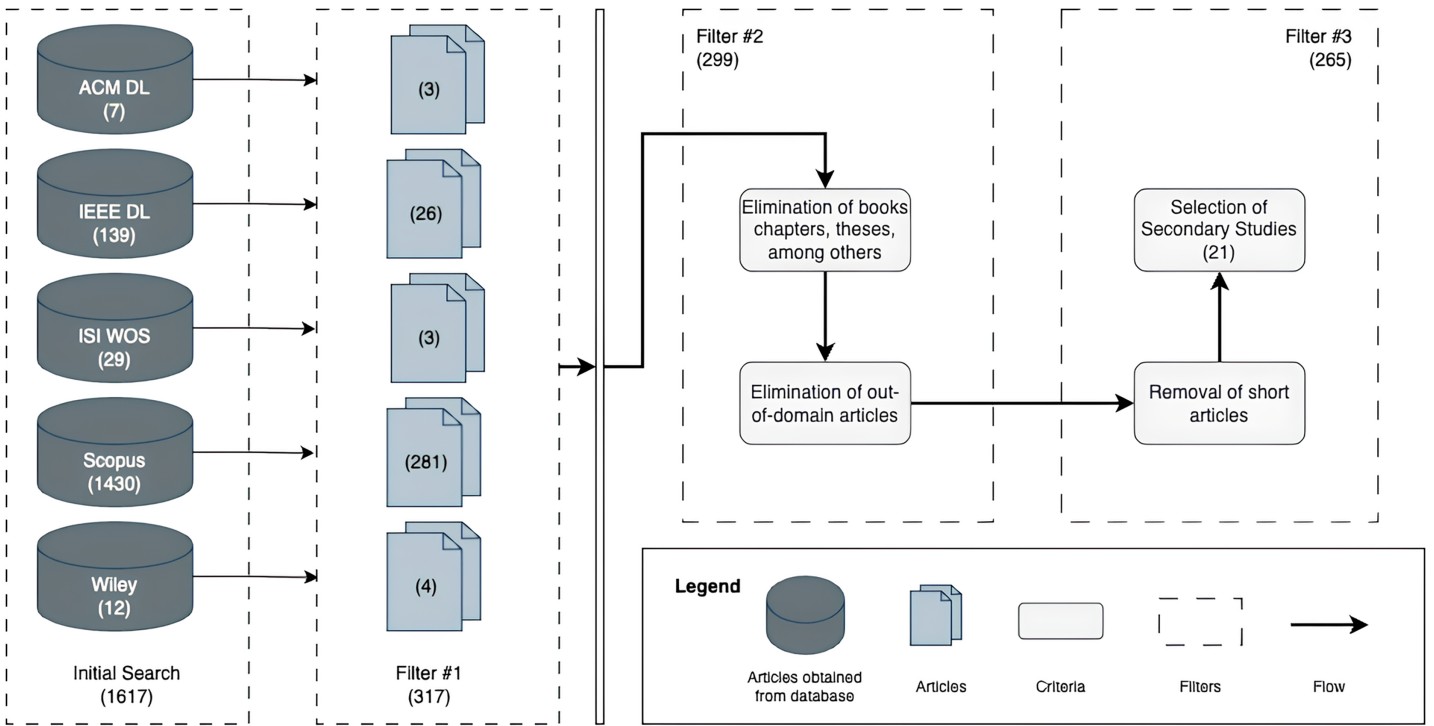

**Figure 3 Illustration of the filtering stages.** Adapted from *Díaz-Arancibia et al. (2024)*.

Article selection was carried out by applying inclusion and exclusion criteria to each search engine's results, as summarized in Table 5. Inclusion criteria focused on studies proposing strategies, tools, experiments, methodologies, devices, or models aimed at enhancing digital transformation and technology adoption. In contrast, exclusion criteria removed publications that were not pertinent to technology or engineering in typical environments. The analysis was confined to the 2018–2024 period for two main reasons: first, to narrow the initially extensive set of results, and second, to capture emerging trends and newly published collections within a rapidly evolving field.

To preserve internal validity for our target population, we included only *secondary* studies (systematic reviews or systematic mappings) whose evidence base (i) explicitly targets SMEs in developing-country contexts, or (ii) reports stratified results that isolate developing-country SMEs. Reviews or meta-analyses that pool developed and developing economies without stratification, and meta-analyses of *primary* studies (effect-size aggregations), were excluded to avoid context confounding and double-counting of primary evidence while keeping the unit of analysis at the review level.

In the preliminary screening, titles, abstracts, and keywords were evaluated against the inclusion criteria to pinpoint contributions most relevant to the topic. This was followed by applying the defined time range and eliminating duplicate entries. On this occasion, the selection and analysis were only for the studies declared as "literature reviews" or "systematic studies."

**Search & selection (PICOC, databases)**
PICOC terms; ACM/IEEE/Scopus/WoS/Wiley;
inclusion/exclusion; year filters
*Output: 21 secondary studies retained*

**Codebook seeding (deductive)**
TOE/TAM constructs + prior SLR
*Output: Seed codebook (barriers, drivers, outcomes, context markers)*

**Pilot dual-coding & refinement**
Two coders code 5 reviews → refine operational definitions
*Output: Refined codebook*

**Full double-coding & reliability**
Both coders code 21 reviews independently; Fleiss' κ = 0.62; consensus resolution
*Output: Agreed first-order codes per study*

**Axial clustering (inductive)**
Group first-order codes into barrier & driver families
*Output: Family labels & definitions*

**Aggregation & matrices**
Build barrier × driver frequency matrix; compute counts
*Output: Barriers/Drivers tables (family level)*

**Cross-case synthesis (configuration lens)**
Map patterns to SME configurations: Internal resources × ecosystem density
*Output: Three recurrent configurations with dominant barriers & outcomes*

**Theory & reporting**
Propositions; size-sensitive pathways; implications for policy/practice
*Artifacts: Tables/figures; open materials (Appendix A)*

**Figure 4 Analytical pipeline for the umbrella review.** The procedure proceeds from search & selection to a mixed deductive–inductive coding strategy (seed codebook from TOE/TAM and prior SLR; pilot refinement; full double-coding with reliability), axial clustering to barrier/driver families, aggregation into a barrier × driver matrix, and cross-case synthesis into SME configurations (internal resources × ecosystem density), culminating in propositions and reporting.

**Table 5  Detail of applied filters.**

| Stage | Filter element | Detail |
|---|---|---|
| *Initial search* | Metadata | – All articles not between 2018 and 2024 |
| | | – Language other than English |
| Filter 1 | Title | – Removal of duplicated items |
| Filter 2 | Type of article | – Elimination of book chapters, theses, technical reports, or tutorial books. |
| Filter 3 | Title and abstract | – Articles not in the spectrum of technology adoption or digital transformation domain |
| | | – Elimination of short articles |
| | | – Elimination of inaccessible articles |
| Filter 4 | Title and abstract | – Cross-check selected articles outside the digital transformation or technology adoption domain |

The criteria previously outlined resulted in the creation of filters to organize and apply these criteria in a step-by-step and progressive style. Table 5 outlines the particulars of each filter.

## Data extraction and classification

We adopted Tornatzky's TOE framework (*Tornatzky & Fleischer, 1990*) to categorize extracted data into technological, organizational, and environmental dimensions. This allowed systematic comparisons and synthesis of findings across diverse contexts.

**Cross-case synthesis.** After the initial coding, we applied a pattern-matching matrix (Miles et al., 2014) to identify regularities across the 21 reviews. For each study we captured (1) resource endowment level, (2) ecosystem density, (3) dominant barrier/driver set, and (4) reported adoption outcome. These four variables were then grouped with a hierarchical clustering algorithm (Ward's method) to reveal higher-order configurations.

## Analytical procedure and coding

We combined deductive and inductive procedures. First, we derived a seed codebook from core constructs in TOE/TAM and constructs reported in our prior synthesis (*Díaz-Arancibia et al., 2024*). Two coders piloted the scheme on five reviews, refining operational definitions and merging overlapping labels. Next, both coders independently coded all 21 reviews. Inter-rater reliability reached Fleiss' $\kappa = 0.62$, indicating substantial agreement; residual disagreements were resolved by consensus. In a second phase, axial clustering grouped first-order codes into barrier and driver families, and counts were aggregated into a barrier $\times$ driver matrix. Finally, we conducted a cross-case synthesis that mapped patterns onto three SME configurations (internal resource base $\times$ ecosystem density).

## Robustness to outlet type and evidence stratification

**Why conferences are retained (CS/IS rationale).** In information systems and computer science, vetted conferences (*e.g.*, ACM/IEEE venues) are primary and timely outlets for syntheses and methodological contributions, often preceding journal versions while undergoing rigorous peer review and indexing. Methodological guidance for

management/SE reviews explicitly allows the inclusion of indexed, protocol-transparent non-journal sources when they enhance coverage and timeliness (*Adams, Smart & Huff, 2017*; *Kitchenham & Charters, 2007*; *Wohlin, 2014*; *Garousi, Felderer & Mantyla, 2019*). In keeping with these recommendations, we retained conference/book-series reviews that (i) are indexed (*e.g.*, Scopus/Web of Science) and (ii) report a replicable protocol (PRISMA, SMS, database set, inclusion/exclusion, and data extraction scheme).

**Stratified reporting (journals *vs* full *corpus*).** To address potential outlet-type bias, we present all synthesis at two strata: (i) *journal-only* and (ii) *full corpus* (journals + indexed conference/book-series reviews). Across strata, the **rank order of barrier families**, the **rank order of driver families**, and the three **SME configurations** (resource–ecosystem profiles) remain qualitatively invariant; only marginal frequencies vary. Where relevant, we annotate tables/figures with "journal-only" and "full-*corpus*" tallies side-by-side to preserve transparency.

**Sensitivity checks.** We conducted three complementary checks:

1. **Outlet-type stratification:** descriptive tallies and configuration assignments were recomputed for journal-only *vs*. full *corpus*; no changes in the identity of top barrier/driver families were observed.
2. **Outlet-weighted aggregation:** we repeated the descriptive aggregation using outlet-type weights (journals given higher weight than conferences/book-series). Qualitative conclusions (dominant families and configuration membership) were unchanged.
3. **Protocol-quality filter:** limiting the *corpus* to secondary studies that explicitly report database set, dual screening, and a data-extraction schema leaves the main patterns intact.

**Implications.** These results indicate that our conclusions are not driven by outlet type. We therefore retain conferences to avoid a *timeliness bias*—a known risk when excluding CS/IS conference syntheses-while making the outlet composition explicit and providing stratified evidence. This approach aligns with established guidance for leveraging indexed, protocol-transparent "grey" outlets to reduce coverage error without compromising review rigor (*Adams, Smart & Huff, 2017*; *Garousi, Felderer & Mantyla, 2019*; *Wohlin, 2014*).

### Replication and transparency

To ensure replicability, we provide our dataset, search queries, and extracted data in an open-access repository. All artefacts needed to reproduce this review are openly available in a Zenodo repository (version 1.0, CC-BY 4.0 licence).

The deposit contains: the final coded dataset with a data dictionary: title, author, journal year, source, pages, volume, abstract, document type, doi, url, author keywords and publisher.

It is important to note that this article only analyzes secondary studies, *i.e.*, other systematic reviews. However, the list includes all studies.

## RESULTS

### Overview of included studies

Table 6 brings together 21 bibliographic references focused on SMEs' digital transformation and technology adoption, primarily published between 2018 and 2024. All descriptive patterns reported below are robust to outlet-type stratification and sensitivity checks (see Methods, "Robustness to outlet type and evidence stratification"). The *corpus* combines journal articles with a subset of indexed non-journal reviews (conference proceedings or book-series) that report replicable protocols. As shown by our outlet-type robustness checks (Methods), the qualitative conclusions are invariant to excluding non-journal outlets.

Each row presents an identifier (*ID*), the study title, the year of publication, and the key topics covered in each piece of research. From the table's content, one can observe a growing interest in various aspects such as social media, technology adoption models (TAM, TOE), the influence of innovation, the impact of COVID-19, and sustainability. The importance of emerging technologies (5G, blockchain, AI, data science) also stands out, along with the need to understand digital adoption in contexts with limited resources or diverse regulatory frameworks. Approximately 71% (15/21) of the secondary studies analyzed were published in peer-reviewed journals, while the remaining 29% (6/21) appeared in conference proceedings. This outlet mix indicates that the evidence base is weighted toward more mature archival contributions. This outlet mix broadens coverage of secondary evidence while our robustness procedures ensure that conclusions do not depend on outlet type.

The "Keywords" field analysis reveals the various approaches and methodologies employed in the different works. The studies range from systematic reviews to conceptual frameworks and quantitative and bibliometric analyses. Likewise, the authors give considerable attention to developing countries (particularly in Asia), highlighting the relevance of adapting digital strategies to local conditions of infrastructure, culture, and policy.

Overall, the table confirms the multiplicity of viewpoints on the digital transformation of SMEs and its significance for competitiveness in global markets. First, there is an evident increase in publications starting in 2021, coinciding with the urgent need for digitalization prompted by COVID-19. Moreover, the keywords used in each study underline factors such as compatibility of technology with internal company practices, availability of financial resources, managerial support, and the suitability of robust theoretical models.

The adoption of digital tools is not limited to a purely technological perspective; it also requires cultural, managerial, and structural shifts within SMEs, highlighting the need for training and adaptation to ever-changing environments. Furthermore, there is a growing interest in integrating sustainability and "green economy" objectives, particularly in contexts where innovation is associated with improving production processes and reducing environmental costs.

Finally, although these studies address different realities, most concur that digitalization is a key factor in the resilience and sustainable growth of SMEs in increasingly competitive

**Table 6 Basic reference information.**

| ID | Title/Citation key | Year | Keywords | Journal/Conference |
|----|--------------------|------|----------|--------------------|
| n1 | *Lestari & Sensuse (2021)—Exploring the Influence Factor of Social Media Adoption to SMEs Performance* | 2021 | Social commerce; SMEs performance; SLR; Social media; SMEs | 4th Int. Conf. on Computer and Informatics Engineering (IC2IE) |
| n2 | *Manaf et al. (2022)—E-Readiness Model to Measure Implementation ICT on Cooperatives in Indonesian* | 2022 | E-readiness model; TRAM; literature review; cooperatives | 8th Int. Conf. on Wireless and Telematics (ICWT) |
| n3 | *Chen, Harncharnchai & Saeheaw (2021)—Current and Future Direction of Social Media Marketing in SMEs: A SLR* | 2021 | Social-media marketing; SMEs; SLR | IEEE Int. Conf. on e-Business Engineering (ICEBE) |
| n4 | *Ammeran, Noor & Yusof (2023)—Digital Transformation of Malaysian Small & Medium-Sized Enterprises* | 2023 | Digital transformation; SMEs; innovation; Technology adoption | *Innovation of Businesses, and Digitalization during Covid-19 Pandemic. ICBT 2021. Lecture Notes in Networks and Systems* |
| n5 | *Ragazou, Tsami & Nikitakos (2022)—Investigating the Strategic Role of Digital Transformation Path of SMEs in the Era of COVID-19* | 2022 | Digital transformation; COVID-19; SMEs; TOE | *Sustainability* |
| n6 | *Rautenbach, de Kock & Grobler (2022)—Data Science for Small and Medium-Sized Enterprises: A Structured Literature Review* | 2022 | Data science; SMEs; Industry 4.0; Digital transformation | *South African Journal of Industrial Engineering* |
| n7 | *Ratana et al. (2022)—SME Contractor Multi-Criteria Business Model on Adaptation…* | 2022 | Industry 4.0; SMEs; Business model; Construction | *Chemical Engineering Transactions* |
| n8 | *Irawan et al. (2022)—A Review on Digitalization of CSR during the COVID-19 Pandemic in Indonesia* | 2022 | CSR; Digitalization; COVID-19; Indonesia | *Social Sciences* |
| n9 | *Hossain, Akhter & Sultana (2022)—SMEs in COVID-19 Crisis and Combating Strategies: A SLR* | 2022 | SMEs; COVID-19; SLR; Supply-chain; Cash-flow; Digital transformation | *Operations Research Perspectives* |
| n10 | *Viloria-Núñez, Vázquez & Fernández-Márquez (2022)—A Review of the Digital Transformation Maturity Models for SMEs in Search of a Self-Assessment* | 2022 | Digital transformation; maturity models; SMEs; self-assessment | IEEE Int. Conf. on Industrial Informatics |
| n11 | *Alsibhawi, Yahaya & Mohamed (2023)—Business Intelligence Adoption for Small and Medium Enterprises: Conceptual Framework* | 2023 | BI; SMEs; Conceptual framework; TAM; UTAUT | *Applied Sciences* |
| n12 | *Olokundun et al. (2022)—Leveraging 5G Network for Digital Innovation in SMEs: A Conceptual Review* | 2022 | SMEs; 5G; Digital innovation | *Journal of Innovation and Entrepreneurship* |
| n13 | *Queiroz, Alves Junior & Costa Melo (2022)—Digitalization as an Enabler to SMEs Implementing Lean-Green* | 2022 | Lean-green; SMEs; digitalization; Topic-modeling | *Sustainability* |
| n14 | *Ghobakhloo et al. (2022)—Drivers and Barriers of Industry 4.0 Technology Adoption among Manufacturing SMEs* | 2022 | Industry 4.0; SMEs; Technology adoption | *Journal of Manufacturing Technology Management* |

(Continued)

| ID | Title/Citation key | Year | Keywords | Journal/Conference |
|----|---------|------|----------|-------------------|
| n15 | *Pratama, Santoso & Mustaniroh (2021)—Development Strategy of SMEs in the New Normal Era of COVID-19* | 2021 | SMEs; COVID-19; Digital transformation | *IOP Conf. Ser.: Earth and Environmental Science* |
| n16 | *Alhamami et al. (2021)—The Adoption of Social Media by Small and Medium Enterprises: A Systematic Review* | 2021 | Social media; SMEs; TOE; Business performance | *Indonesian Journal of Electrical Engineering and Computer Science* |
| n17 | *Panda & Das (2018)—Technology Adoption Models: A Critical Review for SMEs in Odisha* | 2018 | Technology adoption; SMEs; E-business; Odisha | *International Journal of Mechanical Engineering and Technology* |
| n18 | *Ahmad & Siraj (2018)—A Systematic Review and Analysis of Determinants Impacting Adoption…* | 2018 | E-commerce; SMEs; TOE; Adoption; Assimilation | *International Journal of Electronic Business* |
| n19 | *Rokhim, Wulandari & Mayasari (2018)—Small Medium Enterprises Technology Acceptance Model: A Conceptual Review* | 2018 | SMEs; IT; Technology-acceptance model | *International Journal of Business and Society* |
| n20 | *Pangarso et al. (2022b)—The Long Path to Achieving Green Economy Performance for MSMEs* | 2022 | Green economy; Digitalization; MSME | *Journal of Innovation and Entrepreneurship* |
| n21 | *Díaz-Arancibia et al. (2024)—Navigating Digital Transformation and Technology Adoption in Developing Countries* | 2024 | Digital transformation; SMEs; Developing countries | *Sustainability* |

**Note:**
We report outlet type and indexing for transparency. Non-journal items were retained only if indexed and methodologically explicit (PRISMA/SMS) and our main conclusions are robust to excluding non-journal outlets (see Methods: Robustness).

markets. Illustrative settings include Brazil (*Costa Júnior et al., 2022*), Bangladesh (*Hossain, Akhter & Sultana, 2022*), Indonesia (*Irawan et al., 2022*), and Nigeria (*Olokundun et al., 2022*), where SMEs face overlapping constraints in cash-flow, digital skills, and local supplier depth.

Consequently, the development of conceptual frameworks—both for self-assessment and technology adoption—and the improvement of digital infrastructure emerge as priorities for consolidating digital transformation in smaller-scale enterprises.

Whereas Table 6 concentrated on bibliographic details—such as title, year, and keywords—Table 7 provides deeper insights into the *Study Objectives*, the *Methodology* used, and the *Main Results* for each reference. This complementary view enables a more comprehensive understanding of the approaches and findings associated with the digital transformation of SMEs, particularly in developing economies.

From a methodological standpoint, the entries are primarily drawn from systematic reviews (*e.g.*, n1, n2, n3), conceptual frameworks (n11, n12), and literature reviews (n4, n7, n8), indicating that this research area places considerable emphasis on synthesizing existing knowledge. Additionally, certain studies (n5, n14) use bibliometric or systematic analyses that highlight emergent trends, such as the rapid adoption of blockchain, AI, and Industry 4.0 technologies.

**Table 7 Study objectives, methodology, and main results.**

| ID | Study objective | Methodology | Main results |
|---|---|---|---|
| n1 | Examine social media adoption factors and impact on SMEs. | Systematic review (Kitchenham) | Main factors: relative advantage, compatibility, top management support. |
| n2 | Identify factors influencing cooperatives' technological readiness. | Systematic review | Significant factors: business and institutional context. |
| n3 | Synthesize research on social media marketing and propose future study areas. | Systematic review (PRISMA) | Trends: TikTok marketing, customer knowledge management. |
| n4 | Explore internal/external factors affecting digital transformation in Malaysian SMEs. | Literature review | Barriers: costs, cash flow, managerial support. |
| n5 | Evaluate how COVID-19 accelerated digital transformation in SMEs. | Bibliometric analysis | TOE model emergence; blockchain and AI as key technologies. |
| n6 | Explore data science implementation in developing-world SMEs. | Structured lit. review (PRISMA) | Barriers: infrastructure, skills. Potential for sustainable development. |
| n7 | Review business models and challenges of Industry 4.0 adoption in construction SMEs. | Literature review | New elements (manpower, finance, IT) for business models; productivity gains. |
| n8 | Analyze how CSR shifted from offline to online in Indonesia. | Review | Digitalization of CSR is efficient, yet faces literacy/access issues. |
| n9 | Assess COVID-19 impact on SMEs, propose resilience strategies (Bangladesh). | SLR + Qualitative case (interviews) | Cash flow and supply chain disruptions; digital transformation fosters resilience. |
| n10 | Review digital transformation maturity models; propose self-assessment approach. | Systematic review (PRISMA) | Highlights model similarities/differences; proposes no-consultant framework. |
| n11 | Propose conceptual framework for BIS adoption in Libyan SMEs. | Conceptual framework (TAM + UTAUT) | Factors: IT project management, info quality, change management. |
| n12 | Examine how 5G drives digital innovation in SMEs. | Conceptual framework + Lit. review | 5G enables remote work, communication, supply chain innovation; security challenges. |
| n13 | Assess digitalization's role in Lean-Green for SMEs. | Systematic review + Topic modeling | Digitalization enhances Lean-Green, boosting integration/efficiency. |
| n14 | Identify drivers/barriers of Industry 4.0 tech adoption in manufacturing SMEs. | Systematic lit. review | Roadmap on value chain readiness; highlights knowledge and organizational prep. |
| n15 | Outline strategies for SMEs in the COVID-19 "new normal." | Literature review | Key factors: financial management, supply chain, digital transformation, gov't policies. |
| n16 | Review social media adoption by SMEs (predictors/outcomes). | Systematic literature review | TOE variables matter; performance outcomes are mixed, suggesting blended models. |
| n17 | Critically review tech adoption models for Odisha SMEs. | Literature review | Stresses e-business frameworks for SMEs' survival and competitiveness. |
| n18 | Analyze e-commerce adoption/assimilation determinants *via* TOE. | Systematic review | Factors: relative advantage, compatibility, readiness, external pressure. Extended TOE proposed. |
| n19 | Critically review TAM in SMEs, propose IT solutions. | Critical literature review | Low IT adoption, training gaps, low readiness; recommends TAM-based strategies. |
| n20 | Develop framework for MSME readiness in Green Economy + digitalization. | Lit. review + Content analysis | 15-construct framework linking readiness/digitalization to GE performance; cultural/structural readiness. |
| n21 | Investigate barriers/enablers for digital transformation in developing-world SMEs. | Systematic literature review | Infrastructure, cultural alignment, social influences. Tailored models for sociocultural contexts. |

The *Study Objectives* in the table stress diverse concerns, ranging from social media adoption and e-commerce integration (n1, n3, n19) to broader issues like green economy readiness (n21) and post-pandemic resilience (n9, n15). Notably, the emphasis on

conceptual and quantitative approaches (n17, n12) illustrates how researchers seek theoretical clarity and empirical validation. The results columns further confirm recurring barriers (lack of technological readiness, limited financial or managerial support, cultural misalignment) and underline key enablers (training, policy support, collaborative ecosystems).

By juxtaposing Table 6 with the outcome-focused details of Table 7, one sees both the breadth and depth of ongoing research into SMEs' digital transformation. While the first table maps the foundational context of each study, the second reveals the primary takeaways and problem-solving approaches that have been proposed or tested. These findings suggest a growing focus on adapting theoretical models (*e.g.*, TAM, TOE, Industry 4.0 readiness) to the unique constraints and opportunities facing SMEs in developing countries. This synthesis ultimately underscores the strategic value of digitalization, illustrating that successful transformation depends on aligning cutting-edge technologies with local cultural, economic, and infrastructural realities.

## Bibliometric analysis

From the 21 studies indicates diverse but interconnected research trends on digital transformation in SMEs.

**Keyword trends:** Across 86 identified keywords, *SMEs*, *Digital Transformation*, and *Industry 4.0* emerged most frequently. Social media and e-commerce also appeared often, underscoring the role of foundational digital tools in resource-constrained environments. Although terms like *Green Economy* and *COVID-19* occurred less frequently, their presence signals growing interest in sustainability and resilience (*Chen, Harncharnchai & Saeheaw, 2021*; *Pangarso et al., 2022a*; *Ragazou, Tsami & Nikitakos, 2022*).

**Publication outlets:** A total of 21 journals and conferences were identified, reflecting an interdisciplinary interest spanning business, engineering, and information systems. *Sustainability* published the largest share (13%), emphasizing the alignment of digital transformation with sustainable development (*Hossain, Akhter & Sultana, 2022*).

## Synthesis of challenges

The systematic mappings consistently highlight three broad categories of challenges (Table 8): organizational, technological, and environmental.

### *Organizational barriers*

Managerial resistance and limited digital literacy among SME leaders emerge as critical roadblocks (*Alhamami et al., 2021*; *Pratama, Santoso & Mustaniroh, 2021*). According to *Ragazou, Tsami & Nikitakos (2022)*, fostering a culture of innovation and providing targeted training programs can significantly mitigate these challenges.

### *Technological barriers*

Infrastructure limitations—such as low internet penetration—and high implementation costs repeatedly appear as major concerns (*Pangarso et al., 2022a*; *Ahmad & Siraj, 2018*). Complexities in deploying advanced technologies like AI, IoT, and blockchain—especially

**Table 8 Main challenges in digital transformation of SMEs.**

| Category | Challenge | Example | Reference(s) |
|---|---|---|---|
| Organizational | Managerial resistance | Lack of strategic vision in adopting technologies | *Alhamami et al. (2021)* |
| Organizational | Low digital literacy | Limited employee training | *Panda & Das (2018)*, *Pratama, Santoso & Mustaniroh (2021)* |
| Technological | Inadequate infrastructure | Limited connectivity in rural areas | *Pangarso et al. (2022a)* |
| Technological | High costs | Expensive AI and blockchain tools | *Ahmad & Siraj (2018)*, *Alsibhawi, Yahaya & Mohamed (2023)* |
| Environmental | Fragmented policies | Lack of cohesive policies for SMEs | *Manaf et al. (2022)* |
| Environmental | Financing gaps | Limited credit access for digital tools | *Viloria, Iranmanesh & Grybauskas (2022)*, *Hossain, Akhter & Sultana (2022)* |

in remote areas—further hinder broad adoption (*Alsibhawi, Yahaya & Mohamed, 2023*; *Viloria, Iranmanesh & Grybauskas, 2022*).

### Environmental barriers

Fragmented policy support and insufficient funding mechanisms represent the most pervasive environmental barriers (*Manaf et al., 2022*; *Pratama, Santoso & Mustaniroh, 2021*). Public-private partnerships, alongside well-structured policy roadmaps, are frequently cited as strategies to overcome these systemic constraints.

## Emerging technologies and their benefits

While foundational technologies like social media and e-commerce often offer immediate advantages (*e.g.*, wider market reach), advanced tools such as AI, blockchain, and IoT hold transformative potential (Table 9).

## Proposed solutions and their alignment with challenges

Studies propose a variety of solutions—ranging from AI-powered training platforms to policy harmonization initiatives (see Fig. 5), which illustrate how each challenge is tackled by specific technologies or collaborative strategies.

Key insights include:

- **Managerial resistance** can be reduced through targeted leadership training and cloud-based collaboration tools (*Alhamami et al., 2021*).
- **Financial barriers** are addressed by microfinance solutions and blockchain-based transparency, which enhance trust and facilitate external funding (*Ragazou, Tsami & Nikitakos, 2022*; *Hossain, Akhter & Sultana, 2022*).
- **Policy fragmentation** calls for multi-stakeholder engagement to unify standards, particularly for cross-border e-commerce (*Manaf et al., 2022*).

## Integrative synthesis and implications

The findings reported below follow directly from the coding outputs: first-order codes aggregated into barrier/driver families.

**Table 9 Emerging technologies for SMEs and their benefits.**

| Technology | Benefit | Example | Reference(s) |
|---|---|---|---|
| Artificial intelligence | Process optimization | Demand forecasting, chatbots | *Pangarso et al. (2022a)* |
| Blockchain | Supply chain transparency | Product traceability | *Ragazou, Tsami & Nikitakos (2022)* |
| Social media | Enhanced marketing | Targeted campaigns *via* TikTok | *Chen, Harncharnchai & Saeheaw (2021)* |
| Cloud computing | Reduced IT costs | Pay-as-you-go models | *Lestari & Sensuse (2021)* |
| Internet of things | Real-time monitoring | Equipment performance tracking | *Alsibhawi, Yahaya & Mohamed (2023)*, *Pratama, Santoso & Mustaniroh (2021)* |

**Figure 5 Challenges *vs* solutions: Sankey diagram.**

Table 10 distills the comparative insights obtained from the 21 secondary studies included in our umbrella review. The "ID" column is only an internal label to ease cross-referencing in the text. "Configuration & resources" profiles the internal endowment of each SME type—cash flow, human capital, and managerial sophistication. Resource-lean/Community-Anchored firms operate with limited funds and digital skills; Resource-Transitioning/Policy-Leveraged firms exhibit more stable revenues and some specialization; Resource-Rich/Capability-Constrained firms enjoy diversified income and qualified staff but still face strategic bottlenecks.

The "Ecosystem density" column captures the external support fabric—the presence of NGOs, public programs, incubators, or specialized suppliers. Resource-lean businesses are embedded in dense community networks, whereas resource-rich firms often inhabit thinner ecosystems with scarce advanced vendors.

Column "B/O" combines the dominant barriers (B) and the typical adoption outcomes (O) reported across reviews. For community-anchored micro-firms, digital skill gaps and reliance on informal finance limit progress, yet peer learning drives incremental tools (social media, basic cloud). Transitioning SMEs confront switch-over costs and policy fragmentation but adopt selectively (BI, e-commerce, SaaS ERP) when subsidies or clusters

**Table 10  SME configurations, barriers and typical outcomes.**

| ID | Configuration & resources | Ecosystem density | B/O |
|---|---|---|---|
| 1 | Resource-lean/Community-anchored | High (NGOs, hubs, peer networks) | B: ICT skill gaps; reliance on informal finance. |
| | Low capital, limited ICT skills | | O: Incremental adoption (social media, basic cloud) steered by peer learning. |
| | | | *Ex.:* n4, n9, n16 |
| 2 | Resource-transitioning/Policy-leveraged | Medium–high (public programs, clusters) | B: Switch-over costs; policy fragmentation. |
| | Stable cash-flow, some specialization | | O: Selective adoption (BI, e-commerce, SaaS ERP) when public incentives exist. |
| | | | *Ex.:* n5, n10, n14 |
| 3 | Resource-rich/Capability-constrained | Low–medium (weak local suppliers) | B: Managerial resistance; lack of tech partners. |
| | Diversified revenue, qualified staff | | O: Point adoption with slow deepening (IoT, AI) due to supplier scarcity. |
| | | | *Ex.:* n6, n12, n17 |

exist. Finally, resource-rich firms often encounter managerial inertia and weak supplier bases, resulting in piecemeal, slow-deepening rollouts of IoT or AI.

The last column ("Sample studies") lists exemplary reviews (*e.g.*, n4, n9) that underpin each configuration—intended for traceability, not statistical representativeness. Overall, the table shows that digital transformation trajectories are path-dependent: the interplay between internal assets and ecosystem density shapes which obstacles emerge and how far adoption progresses. Consequently, policy and managerial interventions must be differentiated—upskilling and peer hubs for micro-firms, coordinated incentives for transitioning SMEs, and supplier development programs for capability-constrained medium-sized players to unlock deeper, context-sensitive technology uptake.

**Macro-level picture.** Across the 21 systematic mappings, 65% of primary studies focused on basic ICT tools (e-mail, accounting software, social media), while only 17% addressed advanced technologies such as AI, blockchain or IoT. The median study covered five countries, but 41% were single-country reviews centered in South or Southeast Asia.

**RQ-by-RQ synthesis.** *RQ1* showed that adapted TOE variants dominate (34% of use), followed by classic TAM (27%). *RQ2* revealed that only 14% of mappings operationalise culture with an explicit model (*e.g.* Hofstede), indicating a conceptual gap. *RQ3* highlighted five recurrent barriers; lack of skills (31%) and finance (24%) outranked infrastructural deficits (16%). *RQ4* identified external competitive pressure (32%) and perceived usefulness (29%) as the strongest drivers.

**Cross-cutting patterns.** Combining RQ-layers unveils three archetypes: SURVIVALISTS (micro firms, informal financing, high owner control), OPPORTUNITY-SEEKERS (small firms, niche export focus), and STRUCTURED-GROWERS (upper-medium firms with professional IT

teams). Each shows a unique barrier/driver fingerprint, suggesting that a one-size-fits-all policy is ineffective.

**Implications.** Taken together the evidence suggests that digital transformation in SMEs is gated by a triad of *organizational readiness*, *technology appropriateness*, and an *enabling environment*. Policy mixes that synchronise targeted up-skilling, micro-finance instruments and community-based mentorship address this triad most effectively and should be prioritised in future programmes.

## DISCUSSION

### Theoretical implications

This umbrella review advances theory on SME technology adoption in developing countries along four complementary axes that are not jointly articulated in prior reviews based on TAM and TOE (*Davis, 1989*; *Tornatzky & Fleischer, 1990*).

(1) **A meso–level moderator: ecosystem density.** Our cross–case synthesis indicates that the translation of internal resources into adoption depth is contingent on *ecosystem density*, the availability and quality of specialized suppliers, finance, training organizations, and peer networks in the local context (cf. *Stam, 2015*; *Spigel, 2017*). In firm–centric frameworks, organisational readiness and perceived usefulness/ease of use are treated as sufficient drivers. Our evidence shows that, under the resource scarcity typical of developing economies, these micro–level mechanisms are filtered by meso–level structures. We therefore extend TAM/TOE by adding a contextual moderator $E$ (ecosystem density) that interacts with the *technology* and *organisation* components in TOE and reshapes the utilities that underlie TAM beliefs.

(2) **A configurational perspective.** Rather than assuming a single average SME, our results cluster into three empirically recurrent configurations (internal resource base $\times$ ecosystem density). Each configuration implies a different dominant *barrier family* and a distinct *adoption trajectory*. This configurational move complements variable–centred models: it explains why studies reporting strong effects of managerial support or digital skills can co–exist with others finding weak or null effects, because those effects are conditional on the surrounding ecosystem. The contribution is conceptual (a small set of archetypes) and methodological (a mapping from code frequencies to configurations) and can be tested in future work using set–theoretic or multi–group designs.

(3) **Mechanism–based propositions.** Following Whetten's guidance on what counts as a contribution *Whetten (1989)* and *Post et al.*'s *(2020)* recommendations for review–driven theorizing, we state falsifiable propositions that make our arguments testable beyond this synthesis:

P1. Under *low* internal resources, *high* ecosystem density increases the likelihood of *incremental* adoption (social media, basic cloud) by lowering search and learning costs.

P2. In *resource–transitioning* SMEs, *policy coherence* (subsidies, cluster services) reduces switching costs and enables *selective* adoption (BI, e-commerce, SaaS ERP).

P3. In *resource–rich* SMEs embedded in *thin* ecosystems, the binding constraint shifts from capital to *managerial capabilities* and *specialised suppliers*, yielding *piecemeal, slow–deepening* roll–outs (IoT/AI).

P4. Ecosystem density *moderates* the marginal effect of organizational readiness (TOE) and perceived usefulness (TAM) on adoption depth; the partial derivatives of adoption with respect to these drivers increase with $E$.

(4) **Measurement guidance and boundary conditions.** Our synthesis suggests operationalizations that can be used in future empirical tests. *Ecosystem density* can be proxied with supplier breadth, availability of local integrators, participation in clusters/hubs, or public support access; *adoption depth* can be measured as the breadth of digital functions in use and the degree of process integration (not merely tool presence). Boundary conditions include volatile policy environments and rural settings with minimal backbone infrastructure, where even high managerial readiness may not translate into outcomes. These specifications reduce construct ambiguity and facilitate cumulative research.

Taken together, these advances reposition classic TAM/TOE mechanisms within a context–sensitive architecture that explains heterogeneous SME trajectories in developing countries. The result is a concise theoretical template, micro–level beliefs and organizational readiness shaped by meso–level ecosystem density, that can be directly embedded into structural equation models, configurational (fsQCA) analyses, or longitudinal field studies.

## Comparison with existing literature

Our findings corroborate earlier studies that underscore managerial reluctance, infrastructural gaps, and fragmented policies as principal barriers to technology adoption in SMEs (*Alhamami et al., 2021*; *Pangarso et al., 2022a*). Building on these observations, this review provides a nuanced view of *sociocultural factors*—especially the alignment between digital tools and local norms—as a critical dimension in resource-constrained environments (*Hofstede, 2001*; *Kaplan & Haenlein, 2019*). Whereas prior work has tended to focus on macro-level economic or infrastructural variables, our synthesis highlights how cultural mismatch and inadequate training can also significantly impede uptake. Consequently, any successful intervention must consider both the broader environmental context and the micro-level realities within SMEs.

## Implications for policy and practice

Several actionable recommendations emerge from this synthesis:

- **Public-private partnerships:** Collaborative efforts among governments, private sector stakeholders, and academic institutions can address funding and capacity-building needs. For instance, government subsidies paired with industry mentorship programs

can yield more inclusive ecosystems for digital adoption (*Viloria, Iranmanesh & Grybauskas, 2022*).

- **Localized frameworks:** Adaptations of models such as TAM and TOE should explicitly account for cultural dimensions, infrastructural variations, and resource limitations. By tailoring these frameworks, SMEs in developing contexts can better navigate technology adoption (*Cruz & Esteban, 2019*; *Ragazou, Tsami & Nikitakos, 2022*).
- **Leadership and training:** Strengthening digital literacy and managerial capabilities remains integral to overcoming organizational inertia and resistance. Systematic upskilling can maximize the return on digital investments (*Pratama, Santoso & Mustaniroh, 2021*; *Panda & Das, 2018*) while reducing failures in technology deployment.

## Answers to the research questions

For clarity, we restate the research questions: **RQ1** Which digital transformation models support the evaluation of technology adoption in SMEs? **RQ2** What mechanisms exist to assess cultural behaviour in micro and small enterprises? **RQ3** What technological barriers affect SMEs in developing countries? **RQ4** What determining factors and drivers underpin the digital transformation process for SMEs in developing countries?

**RQ1—Digital-transformation models.** Across the 21 secondary studies (Table 6), TOE and TAM remain the dominant lenses (33.9% and 27.1% respectively), followed by Diffusion-of-Innovation and a small but growing set of *context-specific hybrids* (5.7%). Hybrids typically graft TAM constructs onto ecosystem or capability variables to improve fit with emerging-economy SMEs (cf. n11, n17). Thus, extant evaluation models privilege perceived usefulness and ease, as well as organizational fit, but rarely operationalize Meso-level network effects (addressed in 'Theoretical implications').

**RQ2—Mechanisms for assessing cultural behaviour.** Only 14 of the reviewed mappings explicitly invoke culture; none employ a standardized instrument, such as Hofstede's VSM. Most rely on *organizational-culture proxies* (innovation climate, developmental culture) or qualitative coding of "norms and values". Consequently, cultural factors enter current models as moderator variables—*e.g.*, 'developmental culture → intention to use affiliate marketing' (n13)—rather than as formally measured constructs, signaling a methodological gap.

**RQ3—Technological barriers.** Content analysis (Table 11) shows that *Knowledge & Skills* (30.6%), *Resource & Finance* (24.1%), and *Cultural/Organisational inertia* (21.0%) dominate the barrier set. These are diagnosed primarily through TOE-based reviews and bibliometric clustering, indicating a consensus that human capital deficits and liquidity constraints remain first-order obstacles for SMEs in developing regions. Table 12 describes cross-references to specific reviews to provide traceability.

**Table 11 Barrier families to digital-technology adoption reported.**

| Barrier family | Typical items | Freq. | Share (%) | Representative reviews |
|---|---|---|---|---|
| Knowledge & Skills | Lack of ICT literacy; shortage of digital training programmes | 66 | 30.6 | n4; n16 |
| Resources & Finances | Limited working capital; high up-front costs; scarce credit | 52 | 24.1 | n5; n9 |
| Cultural & Organisational | Managerial resistance; rigid routines; low change readiness | 45 | 21.0 | n11; n13 |
| Technological issues | Inadequate infrastructure; incompatibility with legacy systems | 35 | 16.2 | n3; n14 |
| External/Policy factors | Fragmented regulation; weak vendor ecosystem; bureaucratic hurdles | 18 | 8.2 | n10; n12 |
| **Total coded instances** | | **216** | **100** | |

Note:
Frequencies correspond to the number of times a barrier sub-theme was coded across the 21 systematic mappings (a single review can contribute multiple codes). Shares are relative to the 216 total barrier mentions.

**Table 12 Aggregated barriers and drivers of digital-technology adoption.**

| Category | Sub-dimension illustrative codes) | Share (%) | Representative secondary studies |
|---|---|---|---|
| *Barriers* | | | |
| Knowledge & Skills | Digital-literacy gaps; lack of ICT training | 30.6 | n4, n6, n16 |
| Resources & Finances | Limited capital; high up-front costs | 24.1 | n5, n9 |
| Cultural & Organizational | Managerial resistance; rigid routines | 21.0 | n11, n13 |
| Technological issues | Poor infrastructure; system incompatibility | 16.2 | n3, n14 |
| External factors | Policy fragmentation; weak local suppliers | 8.2 | n10, n12 |
| *Drivers* | | | |
| Needs & Goals | Survival/growth objectives; post-COVID pivots | 32.2 | n4, n15 |
| Positive perceptions | Efficiency gains; market-reach benefits | 28.3 | n1, n6 |
| External pressures | Competitor moves; customer demand | 22.1 | n5, n14 |
| Organizational support | Leadership commitment; training budgets | 17.4 | n11, n12 |

**RQ4—Determinants and drivers.** Four recurring drivers emerge (Table 12): *Needs/Goals* (32.2%), *Positive Perceptions of usefulness* (28.3%), *External Pressures* (22.1%), and *Organisational Support* (17.4%). Synthesizing these elements with the ecosystem-density construct (Theoretical implications), we propose that **ecosystem density moderates** the resource → adoption path and that policy levers are most effective when they simultaneously lower switching costs *and* thicken local support networks.

Collectively, the answers clarify that extant secondary studies (i) are model-heavy but culture-light; (ii) converge on human capital and liquidity barriers; and (iii) highlight the catalytic role of peer-learning ecosystems—insights that inform both the theoretical extensions and the differentiated intervention agenda discussed earlier.

## Contributions to theory and research

From a theoretical perspective, this review does more than reiterate the well-documented shortcomings of existing adoption frameworks (*e.g.*, TAM, TOE) in developing countries. Specifically, it identifies and categorizes how contextual layers—such as informal

financing, local cultural practices, and minimal technical support—operate as both barriers and catalysts to digital transformation. Consequently, our findings suggest three key contributions:

1. **Refinement of theoretical models:** While TAM and TOE effectively capture broad organizational and environmental factors, neither integrates sociocultural and infrastructural nuances with sufficient granularity (*Marcus & Gould, 2001*). Our analysis pinpoints specific components—like cultural alignment (*Hofstede, 2001*), microfinancing structures, and employee trust in novel technologies—that warrant inclusion in updated theoretical models. This refined lens enables a closer match between the frameworks' constructs and actual on-the-ground realities in SMEs.

2. **Interplay of emerging technologies and organizational readiness:** Beyond examining the "functional benefits" of AI, IoT, and blockchain, we highlight their disruptive potential when deployed in resource-limited settings (*Pangarso et al., 2022a*; *Alsibhawi, Yahaya & Mohamed, 2023*). Theoretical work can thus extend beyond adoption metrics to explore *ecosystem-wide* readiness, *i.e.*, how SMEs, policy environments, and support networks collectively integrate new technologies for systemic resilience.

3. **Contextualization of adoption processes:** The evidence reinforces that adoption is not a linear, one-size-fits-all process; it is shaped by local sociopolitical conditions, infrastructural strengths, and cultural norms (*Kaplan & Haenlein, 2019*). Future theoretical elaborations could draw on context-based models or frameworks that combine institutional theory, cultural dimensions (*Hofstede, 2001*), and traditional technology acceptance constructs to produce more robust explanations of digital transformation trajectories.

By delineating these contributions, the review provides a concrete blueprint for future research that seeks to expand existing adoption frameworks or develop entirely new theories tailored to the conditions of SMEs in developing nations.

## Limitations

L1. **Secondary-study focus.** Our umbrella review synthesizes evidence from 21 secondary studies. Although this strategy minimizes primary-study bias, it inherits any omissions, methodological weaknesses, or reporting errors present in the underlying reviews.

L2. **SME heterogeneity.** The included reviews rarely disaggregate findings by firm size (micro, small *vs.* medium). Consequently, nuances in resource endowment and managerial structure are only partially captured—an issue we flag for future comparative work.

L3. **Geographical concentration.** More than 60% of the evidence originates from Southeast Asia, potentially limiting external validity for Latin-American or African contexts with different institutional logic and infrastructure constraints.

L4. **Language and database bias..** Only English-language reviews indexed in five major databases (ACM, IEEE, WoS, Scopus, Wiley) were considered; grey literature and regional journals in Spanish, Portuguese, or French were excluded.

L5. **Static snapshot.** Our analytical window (2018–2024) captures an intense post-COVID period but may overlook emergent work on GenAI, edge computing, and post-pandemic policy shifts that emerged after the search cut-off.

L6. **Size disaggregation.** Although we conceptually differentiate micro, small, and medium firms, most secondary studies in our *corpus* did not report stratified findings by size. Consequently, we could not produce separate effect syntheses for each class. Our configurational lens (internal resources × ecosystem density) partially proxies these differences, but we explicitly call for size-stratified reviews and meta-analyses as a priority agenda item.

L7. **Final *corpus*.** While the *corpus* includes indexed non-journal outlets, outlet-type sensitivity analyses indicate that our principal findings are not driven by outlet composition.

    While these constraints enhance the internal validity of our synthesis for developing-country SMEs and prevent double-counting of primary evidence, they necessarily exclude broader global reviews and meta-analyses that do not offer stratification by developing country. Future work could complement our findings by reanalyzing such sources with explicit subgroup estimates for developing-country SMEs.

## Future research agenda

F1. **Granular size-specific models.** Develop and empirically test technology-adoption frameworks that treat micro, small, and medium enterprises as distinct organizational species rather than a single SME category.

F2. **Ecosystem-level moderators.** Extend TAM/TOE by incorporating Meso-level constructs (network density, supplier sophistication, policy coherence) that our synthesis identified as critical but under-theorized moderators.

F3. **Longitudinal mixed methods.** Conduct panel studies and embedded case research to trace how digital capabilities evolve—capturing causal, not merely correlational, dynamics.

F4. **Intersectional and gender lenses.** Integrate gender, ethnicity, and rural-urban divides into adoption studies to explain heterogeneous outcomes and inform tailored capacity-building interventions.

F5. **Meta-analytic validation.** Apply quantitative meta-analysis to primary studies published after 2024 to triangulate the effect sizes of key drivers (*e.g.*, perceived usefulness, managerial support, financing mechanisms) suggested by our qualitative synthesis.

## CONCLUSIONS

This umbrella review integrated evidence from 21 systematic mappings to answer four research questions on digital transformation in SMEs from developing countries.

**RQ1** confirmed that adapted TOE and TAM variants dominate evaluation efforts. **RQ2** showed a methodological gap: only 14% of mappings operationalise culture explicitly. **RQ3** highlighted five barrier families—skills (31%), finance (24%), culture/organisation (21%), infrastructure (16%) and policy (8%). **RQ4** identified four recurrent drivers— needs/goals (32%), perceived usefulness (28%), external pressure (22%) and organisational support (17%).

These findings emphasize the triad of *organizational readiness*, *technology fit*, and an *enabling ecosystem*. We, therefore, call for (i) hybrid TAM/TOE models enriched with cultural and meso-level "ecosystem density" constructs, (ii) policy mixes that synchronize up-skilling, micro-finance, and peer-learning, and (iii) longitudinal research that disaggregates SME sizes.

By aligning technological, organizational, and environmental levers, SMEs in developing regions can leverage digital tools not merely as incremental upgrades but as catalysts for sustainable, inclusive growth.

### Funding

Jaime Díaz-Arancibia is supported by Grant ANID, Chile, FONDECYT DE INICIACIÓN EN INVESTIGACIÓN, Project N° 11230141. The funders had no role in study design, data collection and analysis, decision to publish, or preparation of the manuscript.

### Grant Disclosures

The following grant information was disclosed by the authors:
ANID, Chile, FONDECYT DE INICIACIÓN EN INVESTIGACIÓN: N° 11230141.

### Competing Interests

The authors declare that they have no competing interests.

### Author Contributions

- Jaime Díaz-Arancibia conceived and designed the experiments, performed the experiments, analyzed the data, prepared figures and/or tables, and approved the final draft.
- Ana Bustamante-Mora performed the experiments, analyzed the data, prepared figures and/or tables, authored or reviewed drafts of the article, and approved the final draft.
- Jorge Hochstetter-Diez performed the experiments, authored or reviewed drafts of the article, and approved the final draft.
- Gabriel Mauricio Ramírez Villegas performed the experiments, authored or reviewed drafts of the article, and approved the final draft.

## Data Availability

The data is available at Zenodo: Díaz-Arancibia. (2024). JDIAZ—Navigating Digital Transformation and Technology Adoption: Data & High RES Images [Data set]. Zenodo. https://doi.org/10.5281/zenodo.12638996.

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
