# Peer review of "Systematic mappings on technology adoption in small and medium-sized enterprises from developing countries: an umbrella review"

_PeerJ Computer Science, doi:10.7717/peerj-cs.3395_

## Round 0.1 · original submission · Major Revisions

· Academic Editor

Major Revisions

Reviewer 1 ·

Basic reporting

The standard of written English in the paper is good and the introduction provides clear guidance on the structure of the paper. In their review, the authors carry out an analysis of 22 recent systematic studies focused on the adoption of digital technologies in SMEs. However, I was surprised that there was no reference in the introduction to some of the key journals dealing with entrepreneurship and small firms – see these papers for example (Kahveci, Avunduk, Daim, & Zaim, 2025; Sinha, Raby, & Salari, 2025).

References
Kahveci, E., Avunduk, Z. B., Daim, T., & Zaim, S. (2025). The role of flexibility, digitalization, and crisis response strategy for SMEs: Case of COVID-19. Journal of Small Business Management, 63(3), 1198-1235. doi:10.1080/00472778.2024.2377671
Sinha, K. K., Raby, S., & Salari, T. (2025). Exploring the scope and depth of digitalisation in times of crisis: Implications for SME resilience. International Small Business Journal: Researching Entrepreneurship, 43(3), 219-245. doi:10.1177/02662426241293000

Experimental design

In the methods section, the authors indicate that they searched for studies dealing with micro firms as well as small and medium-sized enterprises. I feel that there really needs to be some discussion about the differences in access to resources between micro firms and especially large medium-sized enterprises. Firms in the latter category will have a more complex organizational structure, more professional managers and access to a wider range of resources (skills, finance etc).

I thought that the various tables and figures were helpful in explaining how the data were collected and analysed. Although I would have expected to have seen some information on the journals/authors included in Table 6 – there is no clear indication of which papers listed in the bibliography are included in the study. Most of these journals listed by the authors are unknown to me and I would have liked to have some indication of their ‘quality’. Journal ranking acts as a proxy for the quality of published work and lower quality journals will have very different acceptance criteria than leading journal in the field (see for example the journal quality list created by the Chartered Association of Business Schools). In other words, the quality of any overarching review is only as good as the sample of studies on which the authors have drawn.

Validity of the findings

I did think that some sections in the paper were far too short – see 3.6/3/7 and especially 4.6, the summary of findings – this was completely inadequate as far as I am concerned. Also, I was very surprised that the authors did not attempt to answer their research questions. This is a major oversight which really must be addressed if the paper is to be published. A related issue concerns the lack of real clarity about what this study have achieved – presumably if the authors answer their research questions this issue will be resolved. Finally, I return to my issue related to the categorisation of SMEs – I was surprised that there was no attempt to distinguish between micro/small firms and medium-sized firm the adoption of digital technologies. Ar the very least, this should have been identified as a limitation of the existing literature.

Additional comments

Overall, I think this is a worthwhile study and with some additional work, as indicated above, there is no reason why it should not be published. I also think that making an effort to strengthen the links with the existing small firm literature would considerably strengthen this study.

·

Basic reporting

I congratulate the authors on the topic of the article. Although I recognize the value of this review, the manuscript has not yet reached its full potential for publication. I would like to point out the following improvements.

1) The title could be more concise and to the point. Authors can even use the term "umbrella review", which was in fact done in this article. For example: "Systematic mappings on technology adoption in SMEs from developing countries: an umbrella review", or just "Technology adoption in SMEs from developing countries: an umbrella review".

2) Section 2.2.2 Measuring or Defining the ”Gap” does not really delve into a research gap. The authors could delve into the rationale for conducting an umbrella review in more depth.

Experimental design

3) Table 7, n17, shows a quantitative study (survey). This study should be excluded, as it is not a literature review like the others. This makes me question whether the inclusion and exclusion criteria adopted were reliable: why was one empirical quantitative study included and others not? Similarly, some conceptual manuscripts were included. Although conceptual articles are based on previous literature, they do not present a replicable logic of selection and review of articles as in a literature review. Therefore, I believe that the inclusion of such articles goes a little beyond the scope of an umbrella review.

Validity of the findings

4) The results only summarize the selected literature reviews, without proposing new concepts, identifying cause and effect relationships, or other theorizing mechanisms. Unfortunately, there are no effective theoretical or practical contributions from the results, which merely "repeat" what other literature reviews have already indicated. In an umbrella review, it is expected that new knowledge will be constructed from the grouping of information from different reviews in a complementary manner. In other words, it is possible to identify patterns or propose explanations for phenomena that isolated studies have not been able to achieve.

Additional comments

5) I recommend the following readings to help the authors propose effective theoretical contributions:

Whetten, D. A. (1989). What constitutes a theoretical contribution? Academy of Management Review, 14(4), 490-495.

Post, C., Sarala, R., Gatrell, C., & Prescott, J. E. (2020). Advancing theory with review articles. Journal of Management Studies, 57(2), 351-376.

---

## Round 0.2 · Major Revisions

· Academic Editor

Major Revisions

**Language Note:** When preparing your next revision, please ensure that your manuscript is reviewed either by a colleague who is proficient in English and familiar with the subject matter, or by a professional editing service. PeerJ offers language editing services; if you are interested, you may contact us at [email protected] for pricing details. Kindly include your manuscript number and title in your inquiry. – PeerJ Staff

Reviewer 1 ·

Basic reporting

The standard of written English is generally very good.

Experimental design

In terms of carrying out their search of the literature dealing with technology adoption in SMEs, I feel that the authors have been too restricted - needed more focus on European journals.

Validity of the findings

I'm afraid that I think the quality of the findings is limited by the weakness of the literature included in the review.

Additional comments

I feel that the authors need to redo their search of the literature - in doing so, they could eliminate some/all of the conference papers. Alternatively, they could give some justification for inclusion of the various conference papers - see the article I recommend about the 'grey literature.

To some extent, the authors have attempted to deal with my comments in their feedback. Taking Observation 2 (from the first review) about the lack of focus on the distinction between the various firms falling under the SME category. Lines 77-88 draw on the World Bank definition of micro firms + SMEs. However, my point is that micro firms have far less resources than either small or medium-sized enterprises, and in the UK they comprise more than 90% of firms in the SME category (and the vast majority of micro firms are actually sole-traders). I suspect that is not greatly different in Malaysia. Therefore, studies that fail to disaggregate firms based on size are, in my view, likely to provide extremely misleading results.

Providing more detail on the sources of literature used in this paper raises a number of difficult questions (Table 6). First, as the authors acknowledge, 29% are conference papers – while some conferences have a rigorous review process (Academy of Management, for example) – many conferences have a less rigorous approach to paper acceptance. Secondly, none of the journals (except one) are known to me, and I have worked in academia and published widely for more than 40 years. The journal that I am aware of – Sustainability – has in the past (at least) been identified as a ‘predatory journal’. In addition, N4 is described as ‘lecture notes’ – in fact, it is actually a book chapter/conference paper and should be identified correctly. As indicated in my previous review, the quality of any literature review is limited by the quality of papers included in that review.
I note that the authors did not address my query about the ‘quality’ of journals included in their review. However, some argue that it is appropriate to include ‘grey’ literature (Adams, Smart, & Huff, 2017). I feel that this is an issue that should be addressed by the authors before this paper is published. Nevertheless, in my view, inclusion of grey literature is only appropriate if the review includes content from ‘respectable’ journals. Alternatively, the authors could indicate that their study draws on the grey literature – ‘Systematic mapping on technology adoption in SMEs: Examining the grey literature’.

In Lines 109-116, the authors provide more information on the distinction between micro firms and medium-sized firms, but do not mention small firms.

Moving on to the research questions – lines 207-208 (Table 1) – the first RQ is very poorly articulated:
(a) What are the digital transformation models that allow evaluating technological adoption in micro and small companies
Something like this would be more appropriate:
(a) Which digital transformation models support the evaluation of technological adoption in SMEs

On lines 267-270, the authors make the following claim about their review:
This outlet mix indicates that the evidence base is weighted toward more mature archival contributions. However, it still captures a substantial share of cutting-edge insights disseminated through conferences—thereby providing a balanced view of both consolidated and emergent knowledge on SME technology adoption in developing-country contexts.

To suggest this review is based on ‘cutting-edge insights’ is extremely misleading (see my comments about grey literature).

In Section 5.4 (line 453), the authors need to restate the research questions in full – rather than providing a summary statement.

I am not convinced that the authors have carried out a particularly convincing search of the literature – there are a number of recent papers which certainly would be appropriate for inclusion in this study (Chouki, Talea, Okar, & Chroqui, 2020; Santini, Matos, Ladeira, Jardim, & Perin, 2023).

In summary, I think there are still a number of weaknesses in this paper that must be addressed prior to publication.

References
Adams, R. J., Smart, P., & Huff, A. S. (2017). Shades of grey: guidelines for working with the grey literature in systematic reviews for management and organizational studies. International Journal of Management Reviews, 19(4), 432-454.
Chouki, M., Talea, M., Okar, C., & Chroqui, R. (2020). Barriers to information technology adoption within small and medium enterprises: A systematic literature review. International Journal of Innovation and Technology Management, 17(01), 2050007.
Santini, F. d. O., Matos, C. A. d., Ladeira, W. J., Jardim, W. C., & Perin, M. G. (2023). Information technology adoption by small and medium enterprises: A meta-analysis. Journal of Small Business & Entrepreneurship, 35(4), 632-655.

·

Basic reporting

I recognize that the article has advanced a lot compared to the first version. Still, some improvements are needed. Regarding the reality of SMEs from developing countries, the authors draw heavily on European sources. It would be helpful to cite articles by researchers from developing countries that portray how SMEs face a shortage of financial and technological resources, such as this paper from Brazil:
Costa Júnior, J.C. et al. (2022). Managing routines and keeping on track: technology, human cognition and performativity in SMEs. Journal of Manufacturing Technology Management, 33(3), 575-597.

Experimental design

Including only literature reviews in the sample is now better employed. Although the authors present the extracted data in an open-access repository, describing how the analysis was performed, even briefly in the methods section, is essential. How did the authors arrive at their final codes? What analysis method was used? A figure could help illustrate this analysis.

Validity of the findings

Some of the findings seem to fall out of nowhere, with no connection to how they were discovered. They need to be better presented based on the data analysis. For example, the motivation for the ecosystem-level analysis is not previously stated in the article.

Additional comments

The sections are very short and lack greater depth in the proposed concepts and how they differ from other concepts already present in the literature.

---

## Round 0.3 · Minor Revisions

· Academic Editor

Minor Revisions

Thank you for your revised manuscript "Systematic mappings on technology adoption in SMEs from developing countries: an umbrella review". Both reviewers recognize the improvements made and consider the paper suitable for publication.

Reviewer 2 recommends acceptance, while Reviewer 1 supports publication pending minor revisions. These include clarifying terms such as “PICOC” and “barrier/driver families” upon first use, correcting the reference to source N4, and ensuring consistency in terminology (e.g., SME vs. MSE in Table 1).

We therefore invite you to submit a final version addressing these minor issues.

Reviewer 1 ·

Basic reporting

The standard of written English is very good.

Experimental design

The study design is appropriate, and the authors have demonstrated high levels of rigor in explaining how they carried out their analysis.

Validity of the findings

Given the relatively limited scope of their review (21 publications), the authors have demonstrated a number of contributions to theory and practice.

Additional comments

I do not have a great deal to add to my previous comments – the authors have dealt reasonably effectively with most of the issues I raised in my last review. I do have a few minor queries that should be addressed before publication. Although the authors mention the term ‘grey literature’ and include the relevant citation, they do not explain what is meant by the term. My additional comments are as follows:

1. The term PICOC is introduced without a clear definition. The term PICOC is clarified on p.228, but not when it is first used p93.
2. N4 is still described as ‘lecture notes’ with a book chapter in parentheses – as far as I am concerned, this is very misleading! (Lecture Notes is the title of the book series in which the Ammeran et al publication appears). The publication should be cited correctly.
3. In Table 1, the term SME becomes MSE – I presume this is a mistake, as there is no attempt to explain what MSE means.
4. The authors need to explain what they mean by barrier/driver families when the terms are first used in the paper.

The authors have been rigorous in searching, selecting, and evaluating the literature on which this paper is based. They are particularly effective in making use of the various tables to illustrate/confirm the nature of their analysis. I still have some reservations about the nature of the evidence base, but these reservations are not sufficient to stop publication of the paper (given the efforts made by the authors to justify their approach).

·

Basic reporting

The manuscript has improved compared to the previous version. The writing is clear and consistent, and the structure now meets the journal’s standards. The authors incorporated non-European references and clarified contextual aspects related to SMEs in developing countries, which enhances the representativeness of the evidence base.

Experimental design

The study design is now adequately explained. The authors clarified the analytical procedure, coding steps, and inclusion criteria for the umbrella review. The addition of the analytical pipeline figure contributes to methodological transparency. While not an exemplary design, it meets the basic requirements for publication.

Validity of the findings

The main findings are now traceable to the analysis described in the methods. The robustness checks strengthen confidence in the results. Although the theoretical discussion could be further developed, the conclusions are consistent with the evidence presented and the study scope.

Additional comments

Overall, the revised version adequately addresses the main concerns raised in the first and second rounds. The paper now meets the minimum requirements for publication in PeerJ Computer Science, showing sufficient methodological rigor and clarity.

---

## Round 0.4 · accepted · Accept

· Academic Editor

Accept

I think that the current version is suitable for publication.